# Preliminary Findings on CTG Expansion Determination in Different Tissues from Patients with Myotonic Dystrophy Type 1

**DOI:** 10.3390/genes11111321

**Published:** 2020-11-07

**Authors:** Alfonsina Ballester-Lopez, Emma Koehorst, Ian Linares-Pardo, Judit Núñez-Manchón, Miriam Almendrote, Giuseppe Lucente, Andrea Arbex, Carles Puente Alonso, Alejandro Lucia, Darren G. Monckton, Sarah A. Cumming, Guillem Pintos-Morell, Jaume Coll-Cantí, Alba Ramos-Fransi, Alicia Martínez-Piñeiro, Gisela Nogales-Gadea

**Affiliations:** 1Neuromuscular and Neuropediatric Research Group, Institut d’Investigació en Ciències de la Salut Germans Trias i Pujol, Campus Can Ruti, Universitat Autònoma de Barcelona, 08916 Badalona, Barcelona, Spain; aballester@igtp.cat (A.B.-L.); ekoehorst@igtp.cat (E.K.); ilinares@igtp.cat (I.L.-P.); judith3194@gmail.com (J.N.-M.); 2Centre for Biomedical Network Research on Rare Diseases (CIBERER), Instituto de Salud Carlos III, 28029 Madrid, Spain; guillempintos@gmail.com; 3Neuromuscular Pathology Unit, Neurology Service, Neuroscience department, Hospital Universitari Germans Trias i Pujol, 08916 Badalona, Barcelona, Spain; miriam.almendrote@gmail.com (M.A.); glucente@igtp.cat (G.L.); andreaarbex@gmail.com (A.A.); jcollc2@gmail.com (J.C.-C.); aramosfransi@gmail.com (A.R.-F.); aliwonpi@gmail.com (A.M.-P.); 4Servei de Cirugia Ortopèdica i Traumatologia, Unitat de mà i nervi Perifèric, Hospital Universitari Germans Trias i Pujol, 08916 Badalona, Spain; cpuentealonso@gmail.com; 5Facultad de Ciencias de la Actividad física y el Deporte, Universidad Europea, 28670 Madrid, Spain; alejandro.lucia@universidadeuropea.es; 6Instituto de Investigación Hospital 12 de Octubre (i+12), 28041 Madrid, Spain; 7Institute of Molecular, Cell and Systems Biology, College of Medical, Veterinary and Life Sciences, University of Glasgow, Glasgow G126QQ, UK; Darren.Monckton@glasgow.ac.uk (D.G.M.); sarah.cumming@glasgow.ac.uk (S.A.C.); 8Division of Rare Diseases, Vall d’Hebron University Hospital, 08035 Barcelona, Spain

**Keywords:** myotonic dystrophy type 1, somatic instability, CTG expansion, blood, muscle, skin

## Abstract

Myotonic Dystrophy type 1 (DM1) is characterized by a high genetic and clinical variability. Determination of the genetic variability in DM1 might help to determine whether there is an association between CTG (Cytosine-Thymine-Guanine) expansion and the clinical manifestations of this condition. We studied the variability of the CTG expansion (progenitor, mode, and longest allele, respectively, and genetic instability) in three tissues (blood, muscle, and tissue) from eight patients with DM1. We also studied the association of genetic data with the patients’ clinical characteristics. Although genetic instability was confirmed in all the tissues that we studied, our results suggest that CTG expansion is larger in muscle and skin cells compared with peripheral blood leukocytes. While keeping in mind that more research is needed in larger cohorts, we have provided preliminary evidence suggesting that the estimated progenitor CTG size in muscle could be potentially used as an indicator of age of disease onset and muscle function impairment.

## 1. Introduction

Myotonic dystrophy type 1 (DM1) is caused by a CTG (Cytosine-Thymine-Guanine) expansion in the 3′ untranslated region of the dystrophia myotonica-protein kinase (*DMPK*) gene [1]. The CTG expansion is highly unstable, showing size variation both within [2] and between tissues [3,4,5,6]. Genetic instability hinders the establishment of genotype/phenotype correlations in patients with DM1, and most studies assessing CTG expansion have focused solely on blood samples [7,8,9,10]. Here we used small pool polymerase chain reaction (SP-PCR) to study CTG expansion in three different tissues from affected patients. We estimated the progenitor allele, the mode of CTG expansion size and the highest CTG repeat number, as well as the genetic instability of the CTG repeat (i.e., the difference between the progenitor and the mode CTG size) in the different tissues. We also analyzed the potential association between the different CTG measures, on the one hand, and patients’ clinical phenotype, on the other.

## 2. Materials and Methods

This study was approved by the local ethics committee (University Hospital Germans Trias i Pujol, # PI15-009) and was performed in agreement with the Declaration of Helsinki for Human Research. All participants signed an informed consent. The study included eight patients with DM1 and eight controls with no previous family history of neuromuscular disorders (recruited from the traumatology department in whom surgery was needed. DM1 diagnosis was confirmed or discarded with triplet primed-PCR^11^ in all the study participants. Clinical information of DM1 patients was obtained from the medical records and updated in the last visit.

We obtained three different samples from patients and controls: blood, muscle biopsy, and skin biopsy. All samples were obtained at the same time. Blood was collected in EDTA tubes and frozen at −20 °C before DNA extraction. The muscle biopsy was obtained from the left biceps muscle in all individuals except for one patient (P8, vastus lateralis muscle). Skin biopsy was obtained with a 0.5 cm skin punch. Muscle biopsies were frozen immediately and stored at −80 °C before DNA extraction. Before freezing skin biopsies at −80 °C for further DNA extraction, they were first seeded and cultured in plates for 6 days to obtain fibroblasts required for other experiments.

Genomic DNA was isolated from peripheral blood [11]. Genomic DNA was extracted from muscle and skin tissue by homogenization in 100 mM Tris-HCl, pH 7.8, and 5 mM EDTA until these tissues were disaggregated. Thereafter tissues were digested in 20 mg/mL proteinase K and 10% SDS for 16 h at 37 °C, and treated with 5.5 M NaCl, phenol and chloroform isoamyl (1:24) before DNA precipitation with isopropanol. DNA quality and quantity were measured by Qubit Fluorometric (Themo Fisher Scientific; Waltham, MA, USA) and Agilent 4200 TapeStation analysis (Agilent Technologies; Santa Clara, CA, USA).

To measure CTG expansion size, SP-PCR was performed [12]. Briefly, small amounts of input DNA (300 pg) were used with the flanking primers DM-C and DM-DR as previously described [12]. We used custom PCR Master Mix (Thermo Fisher Scientific; Waltham, MA, USA) supplemented with 69 mM 2-mercaptoethanol, and Taq polymerase Thermus aquaticus (Sigma-Aldrich; Gillingham, UK) at 1 unit per 10 µL, supplemented with 5% DMSO and the annealing temperature was 63.5 °C. DNA fragments were resolved by electrophoresis on a 1% agarose gel, followed by Southern blot [13]. The estimated CTG sizes (the progenitor, the mode, and the longest CTG size) in each tissue were determined by comparison against the molecular weight ladder, using GelAnalyzer 19.1 software ((www.gelanalyzer.com) by Istvan Lazar Jr., PhD and Istvan Lazar Sr., PhD, CSc). We studied four replicates of each sample, allowing the analysis of the allele distribution CTG sizes, since the most representative allele sizes that are present in a given sample are shown in the gel (see Appendix A for more information on the different measurements). Genetic instability was calculated by subtracting the progenitor CTG size from the mode CTG size amplified for each sample. This method has been optimized by Prof. Monckton’s group [12].

We used repeated-measures, one-factor (i.e., ‘tissue’) analysis of variance (ANOVA) to compare CTG variables (progenitor, mode, longest repeat length, and somatic instability) across the different tissue samples within each subject. When a main tissue effect was found, post hoc pairwise comparisons (skin vs. blood, skin vs. muscle, and blood vs. muscle) were done with the Bonferroni test. We also determined the relationship between the aforementioned CTG variables and patients’ clinical characteristics with Pearson’s correlations (or Spearman correlations for those data that were not normally distributed, as determined with the Shapiro–Wilk test). The level of significance was set at 0.05 (two-tailed).

## 3. Results

We studied eight patients (six women). The patients’ cohort included five unrelated individuals and three sisters from the same family (P3, P4, and P8). Symptom onset occurred during adulthood in seven patients (with symptoms starting around their fifties in two of them (P2, aged 48 years; P7, 50 years)) whereas in one patient the symptoms started earlier in life (P1, 15 years) (Table 1). One patient (P7) carried previously reported CCG interruptions [14]. All the patients had clinical myotonia, but only two showed a mild impairment in biceps muscle strength (as reflected by a score of 4 in the 0 to 5 Medical Research Council (MRC) scale). Performance in the 6-min walking distance test averaged 377 m (range 251–519). When using the muscular impairment rating scale (MIRS), 25% of patients showed minimal signs of muscular impairment, 50% had distal weakness, and 25% had mild-moderate proximal weakness. Most patients (87.5%) were independent in daily life activities (score of 0–2 on the modified Rankin (mRS) (0 to 6) scale), and only one (P5) had a moderately severe disability (4).

The estimated progenitor, most abundant (mode) and longest CTG expansion size were measured in blood (*n* = 8), muscle (*n* = 7) and skin samples (*n* = 5) (Figure 1). An example of CTG expansion determination is shown in Appendix A. DNA from one muscle biopsy and from three skin biopsies yielded no amplification and therefore precluded CTG sizing. We found no differences across tissues for progenitor CTG (*p* = 0.449 for tissue effect with the repeated-measures one-factor ANOVA), mode size (*p* = 0.247), and genetic instability (*p* = 0.691). By contrast, a significant (*p* = 0.041) tissue effect was found for the longest CTG with significant differences (*p* < 0.05) found for all post hoc pairwise comparisons (thus, mean of the longest CTG size in blood (665 CTGs) < muscle (1110 CTGs) < skin (1408 CTGs). In blood samples, we found the following significant correlations (all *p* < 0.05): blood progenitor CTG size vs. mode allele, r = 0.900 (95% confidence interval 0.536–0.982); mode allele vs. longest allele, r = 0.805 (0.231–0.963); and longest allele vs. progenitor allele, r = 0.861 (0.398–0.975). These results suggested that the progenitor, the mode and the longest expansion size were uniformly distributed, with the CTG tract evenly expanded in blood. For muscle and skin samples, a significant (*p* < 0.05) correlation was only found between the progenitor and mode size (muscle: r = 0.769 (0.037–0.964); skin: r = 0.895 (0.061–0.993)). No correlation was found between tissues. Finally, as P7 carried variant repeats, we compared the results of this patient with those of patients carrying pure repeats to determine whether P7 showed a more stable CTG repeat behavior in any of the tissues, which was not the case (Appendix A).

We further studied the relationship between genetic and clinical data and found a significant (*p* < 0.05) correlation between the progenitor allele found in muscle and both age of disease onset (r = −0.850 (−0.977–0.268)) and the MRC corresponding to the studied muscles (r = −0.932 (−0.992–0.496)) (Figure 2). The CTG mode length in muscle was also correlated with the MRC score for the muscle in question (r = −0.898 (−0.989 to −0.319), *p* < 0.05). By contrast, no significant correlation was found between CTG expansion in blood or skin and the clinical manifestation of the disease.

## 4. Discussion

Our preliminary results suggest that CTG expansions might be in general larger in muscle and skin than in blood. Previous studies have reported that (i) patients’ muscle fibers carry larger expansions than peripheral blood leukocytes [3,4,5], and (ii) patients’ fibroblasts carry larger expansions than peripheral blood lymphocytes [6]. However, no previous study has assessed CTG repeats in blood, muscle and skin cells from the same patients. Furthermore, we studied in depth the CTG expansion size by applying SP-PCR methodology and took into account the genetic instability of CTG expansion (instead of focusing on one single CTG size per tissue using a less accurate southern blot-based analysis). This strategy allowed us to determine that the progenitor and the mode size did not differ significantly across tissues, as opposed to the highest expansion. This finding suggests that the CTG tract is expanding in a different manner in each tissue.

The three tissues presented genetic instability. The complex phenomenon of genetic instability can be produced by numerous mechanisms, including not only DNA repair mechanisms but also DNA replication, transcription, and epigenetic changes. In muscle and skin, a non-dividing cell status coupled with DNA repair mechanisms might play an important role in producing genetic instability. In the case of blood cells, the CTG expansion instability could also be affected by the division status of these cells.

No differences in the CTG stability were found between P7 (who carried variant repeats) and the rest of the patients (who had pure expansion repeats in blood, skin, and muscle). Some authors have shown a stabilizing effect of the variant repeats [15,16], which was not confirmed in P7. Although the case of P7 might be an exceptional one, no conclusions can be really drawn as our study is the first to analyze genetic instability in different tissues of a patient with variant repeats.

When analyzing each tissue independently, we found that all measures (progenitor, mode, and highest CTG) were correlated to each other in blood samples, suggesting that the progenitor CTG size leads the genetic instability of CTG expansion in blood. Conversely, in muscle and skin the progenitor was correlated with the mode but not with the highest CTG, suggesting that the genetic instability of the CTG tract is more random in these tissues, probably due to the longer length of CTG repeats.

We found that the progenitor allele in muscle tissue was the only CTG variable associated with age of disease onset. This finding is not surprising when considering that the muscle is one of the most affected tissues in patients with DM1. In this regard, we studied samples mostly from biceps muscle, whereas disease manifestation might start earlier in the tibialis anterior muscle [17,18]. However, some authors have hypothesized that CTG in muscle tissue could show a stronger correlation with disease severity than CTG determined in blood cells [4,5], for which we were actually able to provide preliminary (‘proof-of-concept’) evidence. Furthermore, our data also suggest that the CTG size in muscle is associated with patients’ muscle impairment (as determined with the MRC scale). These results might reflect a close relationship between CTG expansion in muscle and the degree of functional affectation in this tissue. Although it has been shown that the progenitor size determined in peripheral blood leukocytes of patients with DM1 is also a good indicator of age of disease onset [19], we failed to find a correlation between blood measures and age of symptom onset, maybe owing to the small sample size of our study. Previous studies in larger cohorts of patients with DM1 have in fact reported a close relationship between CTG size in blood and cardiac complications [20] or survival [21].

In conclusion, we found preliminary evidence for the presence of genetic instability in all the patients’ tissues that we studied, yet with muscle and skin cells carrying larger expansions than peripheral blood leukocytes. Although more research is needed in larger cohorts, our preliminary data suggest the estimated progenitor CTG size as determined in muscle tissue is associated with age of disease onset and muscle functional impairment.

## Figures and Tables

**Figure 1 genes-11-01321-f001:**
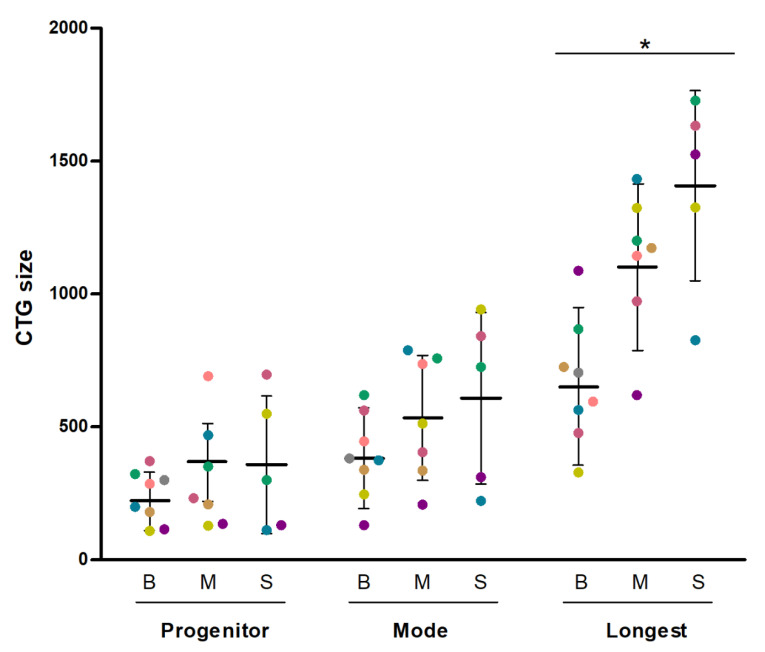
CTG (Cytosine-Thymine-Guanine) repeat number estimates of the progenitor, mode, and longest allele length in the study patients. All the individual CTG data are shown, with each circle representing one single CTG size (using a different color per patient). The mean and SD values of the different CTG measures are also shown. Symbol: * *p* = 0.041 for tissue effect with repeated-measures one-factor (‘tissue’) ANOVA.

**Figure 2 genes-11-01321-f002:**
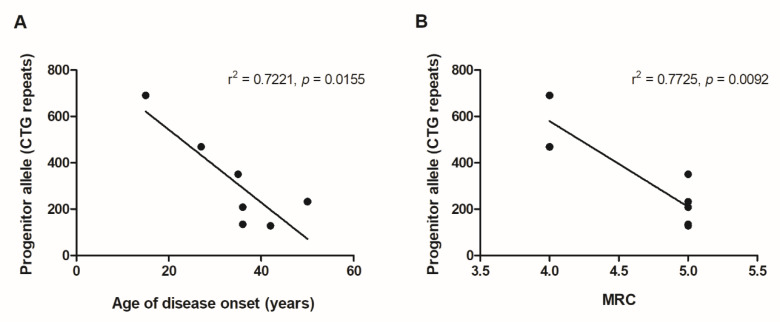
Correlations between the progenitor CTG size present in muscle and the age of disease onset and the MRC of the studied muscles. (**A**) Correlation between the progenitor CTG size present in muscle and the age of disease onset (r = −0.850 (−0.977–−0.268), *p* < 0.05). (**B**) Correlation between the progenitor CTG size present in muscle and the MRC of the studied muscles (r = −0.932 (−0.992–−0.496), *p* < 0.05).

**Table 1 genes-11-01321-t001:** Clinical characteristics of the patients.

Patient	Sex	Age of Symptom Onset (Years)	Age at Sampling (Years)	Biceps Muscle (MRC Scale)	Myotonia (s)	6-min Walking Distance (m)	MIRS	mRS
P1	F	15 *	36	4	0.52	348	4	2
P2	M	48	54	5	0.67	251	3	2
P3	F	36	41	5	0.73	368	2	1
P4	F	42	46	5	0.98	338	3	1
P5	F	27	40	4	NP	NP	4	4
P6	M	36	41	5	0.96	519	3	2
P7	F	50	62	5	NP	436	2	1
P8	F	35	38	5	NP	NP	3	2

Abbreviations: F, female; M, male; MRC, Medical Research Council; NP, not performed. Symbol: * although it was not possible to determine the actual age of disease onset of this patient, since at the first visit (age 36) she had obvious signs that commonly appear early in patient’s life (including oval pallor and temporal atrophy) we considered that the disease onset occurred during adolescence.

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
