# Peer review of "Preliminary Findings on CTG Expansion Determination in Different Tissues from Patients with Myotonic Dystrophy Type 1"

_genes, 2020, doi:10.3390/genes11111321_

Round 1

Reviewer 1 Report

In this manuscript, the authors use small pool PCR to estimate the length of the CTG repeats in myotonic dystrophy type 1 patients. They then compare different CTG repeats lengths (the progenitor, mode and highest CTG length) in three different tissues (blood, muscle and skin) and correlate these measurements to several clinical features like the age of onset or muscle strength or impairment. Given their results, the authors conclude that muscle and skin have larger CTG expansions and more instability than in blood. Furthermore, they propose the progenitor CTG size in muscle as a good indicator of the age of onset and muscle MRC.

This study follows a previous study, from the same authors, defining the small pool PCR (spPCR) method as the best method to measure the CTG repeats.

The strongest point of this study is that the authors have access to three tissues, obtained at the same time, from 8 patients. Their results are not new however, for the first time, it is possible to compare the CTG size (progenitor, mode and longest) in three different tissues of a same patient. The weakest point is that they have a small cohort of patients (and I understand the difficulty to get biopsies from patients) and, as mentioned by the author in the discussion, may affect the findings of significant correlation.

Globally I think that this manuscript lacks details on experiments protocols and analysis methods and may need more in-depth analysis of the results, especially when compared to other studies on CTG instability. Given the small cohort of patients and their results, the authors should also nuance their conclusions and expand some of their sections.

Comments:

- The authors need to better define how they measure the different CTG size in the method section and maybe add some supplementary figures (e.g. representative spPCR gels with indication of each measured length).

- The authors also conclude about the instability observed in their samples but the method to estimate instability is barely indicated in the introduction and the method section (as “difference between the progenitor size and the longest CTG size” which somehow differs from the definition given by D. Monckton’s team). The authors should better define the instability and should include a figure with the instability results. Furthermore, as the instability is an evolution of the CTG size over the lifetime of the patient, are the results more representative of the somatic mosaicism?

- The author should add some figures to illustrate their results, especially when it comes to correlations (e.g. illustration of correlation between progenitor CTG size in muscle and muscle MRC or age of onset could be helpful)

- The authors should give the age of sampling of the patients (as studies have shown that the estimated progenitor length increases during lifetime of the patient) and the estimated CTG size used for the diagnosis.

- The authors need to nuance or clarify some of their sentences

1) in the results section: “all measurements showed larger CTG repeats in muscle and skin compared to blood”. The authors need to change this statement as the only significant results is with the longest CTG group and not in the other measurements (progenitor and mode). Furthermore, I think the author should indicate the p value of the differences observed between tissues in each group (especially in the “longest CTG” group). It is difficult, at least for me, to understand what means the p value associated with the “longest CTG” group.

 2) In the discussion section:

 “ and that the variability of the CTG expansion increases with time” even if it is true, the author do not show any results for this conclusion as they measure the CTG expansion only at one time point.

“However, in our study, the instability in blood and muscle was only found to be correlated with the highest CTG size of each respective tissue”: I was wondering if this correlation is biased because the longest CTG size is the major contributor of the instability as calculated in this study. Maybe the author could discuss a little more about the absence of correlation with the progenitor CTG size and the fact that patients with small progenitor and mode CTG size show very large CTG expansions in muscle

- Do the authors consider the contraction events in their evaluation of instability ?

- The author should discuss if the interruption in the CTG repeats of patient P7 modifies the observed differences in the three tissues

Author Response

Comments appreciated.

We agree that the weakest point of our article is indeed the small sample size of the patients’ cohort. As we have also mentioned to the Editor, obtaining samples from new patients is simply not feasible for us at present, among other reasons because of the invasive nature of biopsy sampling and because the grant in whose frame the study was conducted (with the corresponding Ethics Committee’s approval to take biopsies) has already expired.

However, we have now provided more details on protocols and methods as per the Reviewer’s recommendations. An in-depth analysis of our results has been made, and the conclusions have been nuanced (and ‘toned down’) accordingly. In fact, even the manuscript title has been revised. Overall, although preliminary in essence, we hope that our findings are still valuable in the context of a Brief Report type of article, as they could provide interesting/useful information for other researchers.

Comments:

- The authors need to better define how they measure the different CTG size in the method section and maybe add some supplementary figures (e.g. representative spPCR gels with indication of each measured length).

A more in-depth explanation of the SP-PCR technique has been added in the Methods section (lines 232-237). In addition, a new supplementary Figure 1 is now included to illustrate how we measured progenitor, mode and longest CTG size, respectively.

- The authors also conclude about the instability observed in their samples but the method to estimate instability is barely indicated in the introduction and the method section (as “difference between the progenitor size and the longest CTG size” which somehow differs from the definition given by D. Monckton’s team). The authors should better define the instability and should include a figure with the instability results. Furthermore, as the instability is an evolution of the CTG size over the lifetime of the patient, are the results more representative of the somatic mosaicism?

Comment appreciated and our sincere apologies. We indeed made a mistake in the calculation of instability in the original version of the manuscript. We have now reanalysed our data using the methodology optimized by Prof. Monckton’s team to calculate instability (Prof Monckton is a co-author of our study). Thus, now we only have used the difference of the progenitor allele and the mode allele to do these calculations. With this new analysis, no differences were found among the tissues for genetic instability of the CTG repeat, and the manuscript has been modified/rewritten accordingly. The new method to calculate genetic instability as well as some more information regarding this method have now been added in our methods section (lines 244-246).

Regarding the terminology been used, we apologize because we previously used the terms ‘somatic mosaicism’ and ‘instability’ interchangeably throughout the original version of the manuscript, without considering the subtle differences that in fact exist between both concepts. Indeed, somatic mosaicism refers to the finding of ≥2 genetic sets in the cells of an individual’s tissue that are not transmitted to their progeny; whereas genetic instability refers to a  high frequency of mutations within the genome of a cellular lineage (e.g., due to fragile sites as the CTG repeat in DM1). As we essentially focused on studying the mutations produced in the CTG expansion from its inherited size, we have now used the term ‘genetic instability’ consistently throughout the entire manuscript (instead of ‘mosaicism’).

- The author should add some figures to illustrate their results, especially when it comes to correlations (e.g. illustration of correlation between progenitor CTG size in muscle and muscle MRC or age of onset could be helpful)

Done. Please see new Figure 2.  

- The authors should give the age of sampling of the patients (as studies have shown that the estimated progenitor length increases during lifetime of the patient) and the estimated CTG size used for the diagnosis.

A new column has been added to Table 1 that shows the patients’ age at the moment of sampling. The estimated CTG size used for the diagnosis is not available, since patients were initially diagnosed by triplet-primed PCR and no measurement of the CTG size was done. The first CTG size measure has been done in the context of this article, with the analysis of blood, skin and muscle tissue.

- The authors need to nuance or clarify some of their sentences

1) in the results section: “all measurements showed larger CTG repeats in muscle and skin compared to blood”. The authors need to change this statement as the only significant results is with the longest CTG group and not in the other measurements (progenitor and mode). Furthermore, I think the author should indicate the p value of the differences observed between tissues in each group (especially in the “longest CTG” group). It is difficult, at least for me, to understand what means the p value associated with the “longest CTG” group.

Comments appreciated. This sentence has now been removed from the revised Results section. The Reviewer is indeed quite right – there was no significant difference between muscle, skin or blood samples for the highest CTG value. The mean values for the longest CTG size for blood, muscle and skin are now shown in order to clarify the differences among tissues (lines 93-95). P-values for post hoc pairwise comparisons across the three different tissues for the longest CTG are now shown. We hope that the information is now shown in a clearer way.

 2) In the discussion section:

 “ and that the variability of the CTG expansion increases with time” even if it is true, the author do not show any results for this conclusion as they measure the CTG expansion only at one time point.

We agree. Thanks for catching this in fact. This sentence has now been removed from the Discussion.

“However, in our study, the instability in blood and muscle was only found to be correlated with the highest CTG size of each respective tissue”: I was wondering if this correlation is biased because the longest CTG size is the major contributor of the instability as calculated in this study. Maybe the author could discuss a little more about the absence of correlation with the progenitor CTG size and the fact that patients with small progenitor and mode CTG size show very large CTG expansions in muscle

Thanks for this comment. With the new calculations been done to determine genetic instability, there is no correlation anymore for the longest CTG size. Accordingly, the sentence mentioned by the Reviewer has now been removed from the manuscript.

- Do the authors consider the contraction events in their evaluation of instability?

We calculated instability by subtracting the progenital allele from the mode allele. So, by doing this calculation the contraction events are not considered. This method of calculation has been optimized over the years in Prof. Monckton’s lab. This group originally started by calculating the genetic instability by subtracting the 10th decile from the 90th centile of the bands detected in the SP-PCR (Morales et al. 2012). However, after many years of SP-PCR experiments and hard work with the large OPTIMISTIC patient cohort (Cumming et al. 2019), they have compared the aforementioned method with the one that we used here, and found similar results. Our way of calculating the genetic instability is a simpler method easily applicable to SP-PCR analysis. So, we have relied on the solid expertise of Prof Mockton’s team, and we have not considered the evaluation of contraction events.

- The author should discuss if the interruption in the CTG repeats of patient P7 modifies the observed differences in the three tissues

We did not find a different pattern for P7 compared to the rest of patients in any of the three tissues. To illustrate this point, we have added a new supplementary Figure 2, in which the results of blood tissue SP-PCR in P7 are compared with a pure repeats patient. The typical pattern published by other authors showing a more stable behaviour of the CTG repeat in the SP-PCR was not observed for our patient. Similar results were found for muscle and skin. These new data have now been added to the Results (line 106-109) and Discussion section of the manuscript (156-162), but also emphasizing that they were obtained in only with one patient.

On behalf of all authors, many thanks for this insightful review.

Reviewer 2 Report

The authors report the sizing of CTG in 3 tissues taken on the same time from the same patient. affected by Myotonic Dystrophy type 1:

Although this is si an innovative and original finding ,I have some major concerns:

1- the cohort of patients is indeed too small ( only 8 patients) to draw conclusion;

2 the cohort is not homogeneous:one juvenile, 2 late-onset and 5 adult;

3 the muscle investigated is proximal ( biceps) and less informative of one distal as Tibialis anterior the golden standard muscle to be biopsied( Iachettini et al, EJH 2015, 19:703-709);

#4 -they need to perform more experiments on different muscle (TA)

Author Response

Comments appreciated. 

  • the cohort of patients is indeed too small ( only 8 patients) to draw conclusion;

We agree that our cohort of patients was too small to make strong conclusions. Accordingly, we have now toned down our interpretations and made it clear that these are preliminary findings (starting in fact with the revised title), which we hope are still worth reporting in the format of a Brief report type of article.

On note, obtaining of samples from new patients is simply not feasible for us at present, among other reasons because of the invasive nature of biopsy sampling and because the grant in which frame the study was done (with the corresponding Ethics Committee’s approval to take biopsies) has already expired.

2 the cohort is not homogeneous:one juvenile, 2 late-onset and 5 adult;

Our purpose was not to study a homogeneous patient cohort, among other reasons because clinical heterogeneity is in fact a hallmark of DM1. In fact, the clinical variability of our study cohort is reflective of that found in the department of adult neuromuscular disorders of our Hospital. Our purpose was solely to determine the variation of the CTG repeat in three different tissues obtained at the same time in DM1 patients. In this regard, we don’t think that clinical heterogeneity is a major limitation (on the contrary, it does reflect real-life scenario).

3 the muscle investigated is proximal (biceps) and less informative of one distal as Tibialis anterior the golden standard muscle to be biopsied (Iachettini et al, EJH 2015, 19:703-709);

We agree that tibialis anterior (TA) muscle has been found to be one of the muscles showing earliest abnormalities in patients with DM1. In fact, a study by Coté et al  (Can J Neurol Sci. 2011 Jan;38(1):112-8)  found TA MRI abnormalities in 80% of the patients they studied. Also, thanks for referring us to the Lachettini et al. study, which is now cited in the revised manuscript. Although TA might have been a better choice than biceps muscle in light of these two studies, because DM1 is a genetic disease the CTG expansion is expected to be present in all the cells. As such, the biceps muscle might also provide valid information on the repeat variation.

#4 -they need to perform more experiments on different muscle (TA)

Please see our response above (it is not possible for us to gather new biopsy samples). However, the potential advantage of evaluating CTG repeat size in TA instead of biceps muscle is now mentioned in the revised Discussion (Page 4, lines 175-178).

On behalf of all authors, many thanks for this insightful review.

Reviewer 3 Report

Myotonic dystrophy (DM) type 1 or Steinert disease is a dominantly inherited disorder with a peculiar pattern of multisystemic clinical features affecting skeletal muscle, heart, eye, and endocrine system. In 1992, the mutation responsible for DM1 was identified as a CTG expansion located in the 3' untranslated region of the dystrophia myotonica-protein kinase gene (DMPK). How this untranslated CTG expansion causes DM1 has been a matter of controversy. Recent reports suggest that the clinical features of the disease are caused by a gain of function RNA mechanism in which the CUG repeats alter cellular function, including alternative splicing of various genes.

Attempts to correlate CTG repeat length with progressive DM1 phenotypes, such as muscle power, respiratory and cardiac involvement, have proven difficult and reports in literature are controversial [Chong Nguyen et al., Circ Cardiovasc Gene. 2017 Jun;10(3):e001526; Groh WJ, et al. Muscle Nerve. 2011 May;43(5):648-51].

The AA determine for the first time the repeat expansion length in blood, muscle and skin samples obtained simultaneously in 8 DM1 patients, by using SP-PCR methodology, which permit to investigate the progenitor, the mode and the highest CTG expanding allele.

They show that 1) muscle and skin have larger expansions and are more unstable than blood; 2) the instability of the repeat size is tissue specific and 3) the progenitor allele in muscle is a good indicator of age of disease onset and correlates with muscle function.

The results are very interesting from a scientific point of view and explain why in several reports the severity of clinical features (such as respiratory or cardiac involvement) do not correlate precisely with the size of expansion. However, the small number of patients who gave their consent to the study also shows how this methodological approach will be difficult to implement in clinical practice in other tissues such as heart or respiratory muscles, due to their greater difficulty of access.

The paper is well written and references updated. I suggest to the Authors only to comment the above mentioned papers in the discussion and to add them in references.

Author Response

Comments much appreciated. We have now cited the papers mentioned by the Reviewer in the Discussion section (lines 192-194).

Round 2

Reviewer 1 Report

The authors have edited their manuscript based on reviewers comments and included some new figures to clarify their message. They nuanced their conclusions, add some needed details in their method and answered my questions.

I have still some minor comments:

- The column with the age of sampling (table 1) was not in the version of the manuscript I received. These data have to be included in the final table.

-The authors have to clarify, in the method section, about the processing of the skin samples and the DNA extraction. In this section, they mentioned that the skin biopsies are cultured for 6 days before -80°C freezing. I don’t understand… do you extract DNA from cultured cells or biopsies?

At the end, I think that these results are suitable for publication. I know that obtaining sample from patient is difficult and SP-PCR is a lot of work. However, their preliminary results confirm preliminary results already published and accepted by the DM community during the last 20 years and I do feel that these results are not original enough in the DM1 field.

Author Response

Comments appreciated.  

Comments:

- The column with the age of sampling (table 1) was not in the version of the manuscript I received. These data have to be included in the final table.

Many thanks for noticing. The column with the age at sampling has been now added to the table 1 of the manuscript.

-The authors have to clarify, in the method section, about the processing of the skin samples and the DNA extraction. In this section, they mentioned that the skin biopsies are cultured for 6 days before -80°C freezing. I don’t understand… do you extract DNA from cultured cells or biopsies?

Comment appreciated. Although this information is not directly related with our study, skin biopsies were first used to isolate fibroblast that were required for other experiments. After that process (culture in plates for 6 days), the skin biopsies were frozen at -80ºC for DNA extraction required for our study. We have now modified the information in the method section (see lines 234-237) to clarify this.

On behalf of all authors, many thanks for your review.

Reviewer 2 Report

The authors have addressed all points raised by Referees.

Author Response

On behalf of all authors, thank you so much.